

# Stem-loop structure preference for site-specific RNA editing by APOBEC3A and APOBEC3G

Shraddha Sharma[1] and Bora E. Baysal[1]

Department of Pathology, Roswell Park Cancer Institute, Buffalo, NY, United States of America

## ABSTRACT

APOBEC3A and APOBEC3G cytidine deaminases inhibit viruses and endogenous retrotransposons. We recently demonstrated the novel cellular C-to-U RNA editing function of APOBEC3A and APOBEC3G. Both enzymes deaminate single-stranded DNAs at multiple TC or CC nucleotide sequences, but edit only a select set of RNAs, often at a single TC or CC nucleotide sequence. To examine the specific site preference for APOBEC3A and -3G-mediated RNA editing, we performed mutagenesis studies of the endogenous cellular RNA substrates of both proteins. We demonstrate that both enzymes prefer RNA substrates that have a predicted stem-loop with the reactive C at the 3′-end of the loop. The size of the loop, the nucleotides immediately 5′ to the target cytosine and stability of the stem have a major impact on the level of RNA editing. Our findings show that both sequence and secondary structure are preferred for RNA editing by APOBEC3A and -3G, and suggest an explanation for substrate and site-specificity of RNA editing by APOBEC3A and -3G enzymes.

## INTRODUCTION

The APOBEC3 (A3) family of cytidine deaminases restricts endogenous retroelements and exogenous viruses and therefore plays an important role in the vertebrate innate immune system (*Cullen, 2006*; *Chiu & Greene, 2008*; *Harris & Dudley, 2015*). Recent studies also suggest that A3 enzymes help retroviruses escape from drugs and adaptive immune recognition (*Monajemi et al., 2012*; *Grant & Larijani, 2017*). The A3 family comprises seven homologous enzymes in primates (*Jarmuz et al., 2002*; *Conticello, 2008*; *Prohaska et al., 2014*) that have either one (A3A, A3C and A3H) or two (A3B, A3D, A3F and A3G) zinc (Zn)-coordinating catalytic domains with $HX_1EX_{23–24}CX_{2–4}C$ motifs (X is any amino acid). The histidine and cysteine residues coordinate $Zn^{2+}$ (*Betts et al., 1994*), and the glutamic acid residue may function as a proton shuttle during the deaminase reaction (*Betts et al., 1994*).

A3 proteins can bind to both ssDNA and ssRNA oligonucleotides (*Prohaska et al., 2014*). However, prior structural and biochemical studies have focused on the interaction of A3 enzymes and ssDNA oligonucleotides since C-to-U (C>U) deamination has been demonstrated in ssDNA exclusively. The A3 family members prefer a thymine immediately 5′ to the target C, except APOBEC3G (A3G), which prefers a cytosine at the 5′ position in

Corresponding authors
Shraddha Sharma,
shraddha.sharma@roswellpark.org
Bora E. Baysal,
Bora.Baysal@roswellpark.org

their ssDNA substrates (*Refsland & Harris, 2013* and references therein). A study by *Mitra et al. (2014)* reported that ssRNA is not a substrate for A3A since ssRNA binds to A3A weakly as compared to ssDNA and A3A-mediated ssRNA deamination was not detected. A3G has been shown to bind to both ssDNA and ssRNA with similar affinities (*Iwatani et al., 2006*). While A3G deaminates ssDNA, no deamination was detected in ssRNA (*Iwatani et al., 2006*)

APOBEC3A (A3A) is highly expressed in monocytes and macrophages and its expression is upregulated on treatment with interferon-α (*Chen et al., 2006*; *Peng et al., 2007*; *Koning et al., 2009*). We recently demonstrated the novel RNA editing function of A3A in monocytes and monocyte-derived macrophages (*Sharma et al., 2015*). A3A induces site-specific RNA editing in mRNAs from hundreds of genes in response to low-oxygen (hypoxia) and interferon type 1 (IFN-1) treatment. Of these edited transcripts, 128 out of 211 and 93 out of 116 edited sites are in the coding exons in monocytes and macrophages, respectively (*Sharma et al., 2015*). On transiently expressing A3A in HEK293T cells, mRNAs of thousands of genes undergo site-specific editing (*Sharma et al., 2017*). Furthermore, we demonstrated site-specific editing of ssRNA with purified recombinant A3A *in vitro*, whereas DNA editing is non-specific and occurs at multiple TC nucleotides (edited C underlined) (*Sharma et al., 2015*). More recently, we have identified the RNA editing function of a second member of the A3 family- the two-domain cytidine deaminase and an anti-HIV-1 restriction factor A3G by transient expression in HEK293T cells (*Sharma et al., 2016*). Interestingly, computational analysis revealed that the edited targets in >70% of A3A substrates in monocytes and macrophages and 95% substrates in 293T cells are flanked by palindromic sequences (*Sharma et al., 2015*; *Sharma et al., 2017*). In the case of A3G, ∼98% of the edited targets are flanked by inverted repeats in 293T cells (*Sharma et al., 2016*). These bioinformatic observations suggested that RNAs with predicted stem-loop structures may be preferentially targeted for RNA editing by A3A and A3G (*Sharma et al., 2015*; *Sharma et al., 2016*; *Sharma et al., 2017*). However, the underlying mechanism for this preference is not clear. To test the hypothesis that RNA stem-loop structure is important for RNA editing by A3A and A3G, we generated a panel of RNA mutants and examined the features of endogenous substrates of A3A and A3G required for RNA editing. Here we experimentally demonstrate for the first time the preference for a stem-loop structure for site-specific A3A and A3G-mediated RNA editing.

## MATERIAL AND METHODS

### Cell culture, plasmids and transfection

Cell cultures of primary monocyte-enriched PBMCs, exposure to hypoxia (1% oxygen) and interferon type 1 were performed as previously described (*Sharma et al., 2015*).

Plasmid constructs for expression of human A3A cDNA, for the generation of C-terminal Myc-DDK-tagged A3A and A3G, pcDNA 3.1(+) vector (used as an empty vector control) were obtained from sources mentioned in *Sharma et al. (2015)* and *Sharma et al. (2016)*.

The TLA-HEK293T human embryonic kidney cells (293T cells) (Open Biosystems, Lafayette, CO, USA) were transfected with plasmid DNA using the jetPRIME (Polyplus-transfection) reagent as per the manufacturer's instructions. The transfection efficiency

was 60%–80% as assessed by fluorescent microscopy of cells that were transfected with the pLemiR plasmid (Open Biosystems, Lafayette, CO, USA) for expression of a red fluorescent protein. Cells were harvested 2 days following transfection.

## Purification of recombinant A3A proteins

The WT A3A was purified as described in *Sharma et al. (2015)*. Briefly, Rosetta 2(DE3)pLysS *E. coli* (EMD Millipore, Burlington, MA, USA) transformed with a bacterial expression construct for C-terminal His$_6$-tagged WT A3A was grown in Luria broth at 37 °C. The cells were induced for expression of the recombinant protein with 0.3 mM isopropyl β-D-1-thiogalactopyranoside and cultured overnight at 18 °C. A3A protein was purified from the lysates by affinity chromatography using the Ni-NTA His bind Resin (EMD Millipore). The concentrated protein was stored in 25 mM Tris (pH 8.0) with 50 mM NaCl, 1 mM DTT, 5% v/v glycerol and 0.02% w/v sodium azide at −80 °C.

## Predicting RNA secondary structures

18 nucleotides (with 7 nucleotides flanking on each side of the tetra-loop sequence) of WT *SDHB*, *TMEM109* and *APP* RNAs were folded using the Mfold nucleic acid folding program (*Zuker, 2003*). 18 nucleotides of WT *PRPSAP2* RNA were folded using both Mfold and RNAfold 2.3.2 (*Zuker, 2003*; *Lorenz et al., 2011*). No optional parameters were used. A single structure along with the minimum free energy value for the structure was obtained for the selected RNAs and is represented in Fig. S1.

## RNA mutagenesis and RNA editing assays

The DNA templates for generating WT and mutant *SDHB* (except M1, M6, M7), *TMEM109* and *APP* RNAs were amplified using oligonucleotide primers listed in Table S1. M1, M6 and M7 *SDHB* RNAs were generated from the 1.1 kb complete SDHB ORF encoding plasmid (RC203182, Origene) following site-directed mutagenesis and *Xho*I linearization of the plasmid DNA. Sanger sequencing was performed on all DNA templates to confirm the desired mutations, which were then *in vitro* transcribed to generate RNAs using reagents and methods provided with the MEGAscript or MEGAshortscript T7 Transcription Kit (Life Technologies, Carlsbad, CA, USA). RNAs isolated from the transcription reaction were treated with DNAse I (Thermo Fisher Scientific, Waltham, MA, USA) and their integrity was verified by electrophoresis on an agarose gel.

In vitro RNA-editing assay with purified APOBEC3A contained 1–10 μM APOBEC3A, 50 pg of synthetic RNAs, 10 mM Tris (pH 8.0), 50 mM KCl and 10 μM ZnCl$_2$. The reactions were incubated for 2 h at 37 °C. RNA was purified from the reactions using TRIzol (Life Technologies, Carlsbad, CA, USA) as per the manufacturer's instructions and reverse transcribed to generate cDNAs as described previously (*Sharma et al., 2015*). The 136C>U editing of the WT and certain *SDHB* RNA mutants (M1, M3–M7) was assessed by allele-specific AS-RT–qPCR as described previously (*Sharma et al., 2015*; *Baysal et al., 2013*), whereas RNA editing levels for remainder of the mutant RNAs along with the WT controls were determined by Sanger sequencing, using the primers listed in Table S1, because these mutants could not be amplified by AS-RT-qPCR reverse primers (Table S2, Data S1). We have previously shown (*Sharma et al., 2015*, Fig. S8) a strong

positive correlation ($r = 0.94$) between *SDHB* 136C>U RNA editing level measurements obtained by AS-RT-qPCR and Sanger sequencing, although estimates obtained by Sanger sequencing were somewhat higher (slope of correlation = 0.71). Thus, it is possible that true editing levels in SDHB constructs M2, M8, M9 and M10, which were estimated by Sanger sequencing (Data S1), might be actually lower (by approximately 30%).

Since *in vitro* RNA editing by A3G has not yet been demonstrated, to examine the impact of RNA mutations in the A3G substrate *PRPSAP2* on RNA editing, we co-transfected mutated PRPSAP2 expression plasmid with A3G expression plasmid in 293T cells. The mutations were performed by site-directed mutagenesis (New England Biolabs, Ipswich, MA, USA) in the PRPSAP2 expression plasmid (clone ID Ohu59963, RefSeq accession XM_011523960; GenScript). Total RNA was isolated and RT-PCR was performed using a *PRPSAP2*-specific forward primer and a vector specific reverse primer complementary to the DDK tag sequence (Table S1). These primers specifically amplified the plasmid derived *PRPSAP2* transcripts but not the endogenously expressed transcripts, allowing us to directly examine the impact of RNA mutations on A3G-mediated RNA editing.

### Estimation of RNA editing levels by Sanger sequencing

Sequencing primers (Integrated DNA Technologies) for the WT and mutant cDNAs generated from RNAs are listed and underlined in Table S1. The PCR products were examined by agarose gel electrophoresis to verify their size and then sequenced on the 3130 xL Genetic Analyzer (Life Technologies, Carlsbad, CA, USA) at the RPCI genomic core facility as described in (*Sharma et al., 2016*). The major and minor chromatogram peak heights at putative edited nucleotides were quantified with Sequencher 5.0/5.1 software (Gene Codes, Ann Arbor, MI) in order to calculate the editing level for the position (Data S1). Since the software identifies a minor peak only if its height is at least 5% that of the major peak's, we have considered $0.048 (= 5/(100 + 5))$ as the detection threshold (*Sharma et al., 2016*; Data S1).

## RESULTS

### Preference for stem-loop structure for site-specific A3A and A3G-mediated RNA editing

Previous studies have shown that A3A-mediated DNA deamination of synthetic oligonucleotides occurs non-specifically at TC dinucleotides (*Chen et al., 2006*; *Shinohara et al., 2012*; *Sharma et al., 2015*; *Chan et al., 2015*). However, A3A-mediated cellular ssRNA editing is site-specific, and bioinformatic analyses predicted that approximately 70% of the edited Cs in A3A's RNA substrates are located within secondary structures (*Sharma et al., 2015*). The most common secondary structure is predicted to be comprised of a CAUC tetra-loop flanked by an average of three palindromic nucleotides (*Sharma et al., 2015*). Similarly, bioinformatics analyses predicted that ~98% of the edited Cs in A3G RNA substrates are located within secondary structures; the most common structure comprising of CNCC (N is any nucleotide) flanked by an average of four palindromic nucleotides (*Sharma et al., 2016*). Separately, while validating edited sites in primary monocytes by Sanger sequence analysis, we observed that a silent A/G single nucleotide polymorphism

(SNP) in the A3A substrate *C1QA* mRNA (rs172378) markedly increased C>U RNA editing three nucleotides upstream of the polymorphism (Fig. 1A). The A>G change in the *C1QA* mRNA (rs172378) is predicted to increase the stem length and subsequently the stem stability of a putative stem-loop structure, resulting in increased RNA editing. While the CCCCCUCGG(a/a) (expressed SNP variation in lower case) sequence shows 11% and 21% editing in 2 donors, CCCCCUCGG(a/g) increased the average editing to 40% when monocyte-enriched peripheral blood mononuclear cells (MEPs) were exposed to hypoxia/IFN-1 (Fig. 1A). Although the estimated RNA editing level may have been affected by background noise (in donor 2 and 3 hypoxia/IFN-1 samples), visual inspection of the chromatograms clearly show that the heterozygous donor 1 cellular RNAs are edited at a higher level compared to homozygous donor 2 and 3 cellular RNAs.

We thus hypothesized that stem-loop RNAs are preferred substrates for editing by APOBEC3A and -3G proteins. We selected three site-specifically edited A3A mRNA substrates-*SDHB* (*NM_003000: c.136C>U, R46X*), *APP* (*NM_001204302: c.1546C>U, R516C*), *TMEM109* (*NM_024092: c.109C>U, R37X*) (*Sharma et al., 2015*; *Sharma et al., 2017*) and one such A3G substrate, *PRPSAP2* (*NM_001243941: c.664C>U, R222W*) (*Sharma et al., 2016*) for further analysis. On analysis of 18 nucleotides of RNA sequence containing the target C by the Mfold (*Zuker, 2003*) or RNAfold (*Lorenz et al., 2011*) nucleic acid folding prediction programs, secondary structures with ΔG values between −5 to −6 kcal/mol are predicted for *SDHB*, *APP* and *PRPSAP2* RNAs (Fig. S1). The predicted secondary structure for these RNAs is a tetra-loop with the edited C at the 3′ end of the loop flanked by a stem containing 3–5 base pairs (bp). *TMEM109* is predicted to form a hepta-loop with a four bp stem and a ΔG value of −1.7 kcal/mol. To test the importance of stem-loop structures for A3A and A3G-mediated RNA editing, we created various mutations (see methods) in the putative loop and stem regions of A3A substrates *SDHB*, *APP* and *TMEM109* and the A3G substrate *PRPSAP2* (Figs. 1B–1D) and assessed their editing levels. *SDHB*, *APP*, *TMEM109* RNAs show ~83%, 24%, 51% site-specific editing in an *in vitro* system, respectively and *PRPSAP2* shows ~44% RNA editing in a cell based system. The RNA editing levels were analysed by AS-RT-qPCR for the *SDHB* mutants, except those which did not have a reverse primer compatible for AS-RT-qPCR analysis of RNA editing (see methods; Table S2). The remainder of the *SDHB*, *APP*, *TMEM109* and *PRPSAP2* mutants were analysed by Sanger sequencing (Data S1). In either method for assessing RNA editing levels, WT RNA substrates were used as a positive control. For convenience in data interpretation, RNA editing of WT substrates is set to 100% and that of mutant RNAs is reported as a fraction of that observed with the WT substrates.

We tested the importance of the −1 nucleotide (nt) (immediately 5′ to C) in A3A and A3G substrates. C>U editing sites are most commonly present within a CCAUCG sequence motif in ssRNA A3A substrates (*Sharma et al., 2015*). Changing the −1 U to C, (UC>CC) in the predicted loop region of the *SDHB* RNA (Fig. 1B, M1), markedly reduced A3A-mediated RNA editing from the normalized value of 100% to 19%. Unlike most A3A substrates, which prefer U at −1 position, in *APP* RNA the −1 nt is occupied by C. Interestingly, substituting C at −1 with U in *APP* RNA (CC>UC), increased editing to 364% (Fig. 1C, M1A). The majority of C>U editing sites in A3G substrates are present

Sharma and Baysal (2017), *PeerJ*, DOI 10.7717/peerj.4136

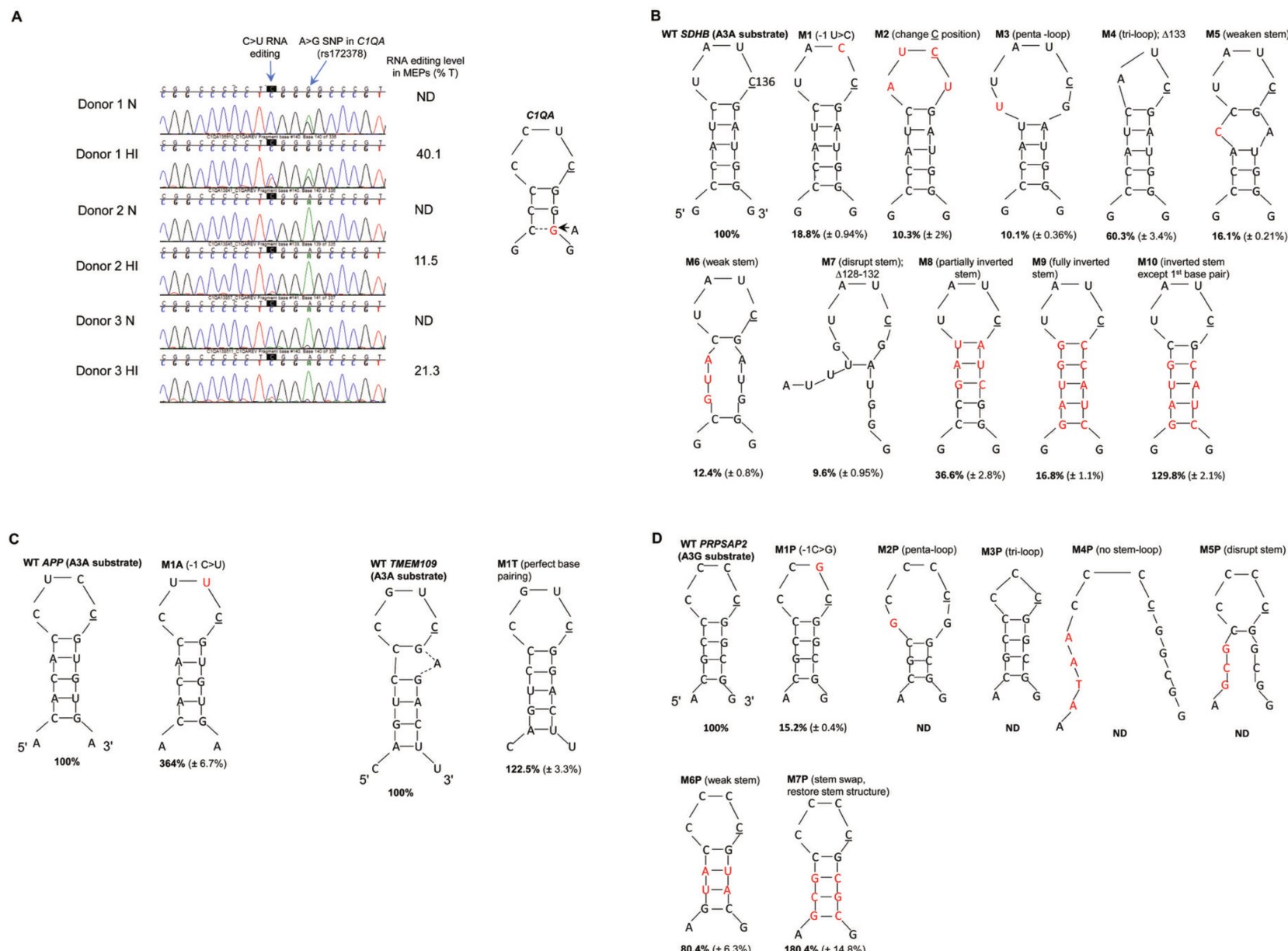

**Figure 1  A3A and A3G prefer predicted stem-loop structures in their RNA substrates.** (A) A3A-mediated RNA editing in normoxia (N) and hypoxia and IFN-1 (HI) treated MEPs of three independent donors. C>T(U) editing is characterized by the emergence of a secondary T peak (red) accompanied by a reduction in height of C peak (blue). A>G silent nucleotide polymorphism (SNP rs172378) in *C1QA* RNA of donor 1 increases C>U editing level as an additional base pair (represented by a dashed line) is predicted to form in the stem of the putative stem-loop. Edited C is underlined. (B) A3A-mediated editing in WT and mutant *SDHB* RNA. 

**Figure 1 (…continued)**
WT *SDHB* RNA forms a putative tetra-loop flanked by a 5 bp stem. Mutations (M) are described above the stem-loop and the mutated nucleotides are colored red in the figure. The average percentage RNA editing of $n = 3$ ($n = 2$ for M1, 6 and 7) is shown in bold and the standard deviations are within parenthesis. The percentage RNA editing in c.136C>U was calculated using allele-specific RT-qPCR (see methods), except M2, M8, M9 and M10 which were calculated using the Sequencer™ 5.0 software (see methods). WT RNA editing was set to 100% and the mutants were calculated as a fraction of the WT. (C) A3A-mediated editing in WT and mutant *APP* and *TMEM109* RNAs. WT *APP* RNA forms a putative tetra-loop flanked by a 5 bp stem. WT *TMEM109* forms a putative tetra-loop flanked by a 5 bp stem and the unpaired adenosine (A) bulges out. (D) A3G-mediated RNA editing of *PRPSAP2* RNA, which forms a putative tetra-loop flanked by a 4 bp stem. For (C) and (D), mutations (M) are described above the stem loop and the mutated/inserted nucleotides are marked in red. The average percentage RNA editing of $n = 3$ is shown in bold and the standard deviations are within parenthesis. The percentage RNA editing was calculated using the Sequencer™ 5.0 software. ND: RNA editing not detectable (below threshold).

within a CNCC [A/G] sequence and therefore prefer C at −1 position (*Sharma et al., 2016*). Changing −1 nt to G (CC>GC) in the A3G substrate *PRPSAP2* RNA loop markedly reduced RNA editing to 15% as compared to WT (Fig. 1D, M1P). These results suggest a preference for U and C at the −1 position in the loop regions of A3A and A3G substrates, respectively.

We next tested the importance of the location of the reactive C within the predicted loop region of the A3A RNA substrate, *SDHB*. As mentioned above, our computational analysis predicts that the edited C is generally located at the 3′ end of the tetra-loop (*Sharma et al., 2015*; *Sharma et al., 2016*; *Sharma et al., 2017*). Changing the position of edited C one nucleotide upstream within the loop in *SDHB* RNA, while maintaining U at −1 position, greatly reduced RNA editing to 10% (Fig. 1B, M2). This result suggests that position of the reactive C within the loop is critical for RNA editing.

The majority of known A3A and A3G RNA substrates are predicted to form a tetra-loop structure (*Sharma et al., 2015*; *Sharma et al., 2016*; *Sharma et al., 2017*). To test whether the size of the loop plays a role in RNA editing, we created substitutions that increase or decrease the predicted loop size in *SDHB* and *PRPSAP2* RNAs (Fig. 1B, M3 and M4; Fig. 1D, M2P and M3P). Increasing from a tetra-loop to a penta-loop (Fig. 1B, M3) reduced RNA editing to 10% in the *SDHB* RNA, and decreasing the size to a tri-loop (Fig. 1B, M4) diminished editing to 60% as compared to the WT *SDHB* RNA. Changing the size of the loop (penta- or tri-loop) of the *PRPSAP2* RNA abolished A3G-mediated RNA editing (Fig. 1D, M2P and M3P). These results suggest that a larger loop is detrimental to both A3A and A3G-mediated RNA editing, whereas reducing the size of the loop to three nucleotides may be tolerated better.

We next tested whether the sequence and/or structure as well as stability of the predicted stem are determinants of RNA editing. We weakened or disrupted the predicted stem by decreasing the number of complementary base pairs in *SDHB* and *PRPSAP2* RNAs (Fig. 1B, M5, M6 and M7; Fig. 1D, M4P and M5P). All of these changes reduced *SDHB* RNA editing 5–10 fold to 16%, 12% and 10%, respectively (Fig. 1B M5, M6 and M7) and abolished A3G-mediated RNA editing of *PRPSAP2* (Fig. 1D, M4P and M5P). Further, altering (inverting/swapping) the sequence of the stem while maintaining base-pairing of the *SDHB* RNA (Fig. 1B, M8 and M9) also reduced RNA editing levels to 37% and 17% as

compared to WT, respectively. These observations suggest that both sequence and stability of the RNA structure are important for optimum RNA editing.

Usually the +1 position (with regard to C) in A3A substrates is occupied by a G base-paired with C or in some cases A, which was substituted with A in M8 (37% editing) and C in M9 (17% editing) (Fig. 1B). Hence, to test the importance of G at +1 position, we created another mutant that retained the first base pair of the predicted stem (G at +1) as WT *SDHB,* but remainder of the stem sequence and structure was similar to the M9 *SDHB* mutant (Fig. 1B, M9 and M10). Changing C at +1 position in the M9 mutant to G in the M10 mutant (Fig. 1B) increased A3A-mediated RNA editing from ∼17% to 130%, respectively (Fig. 1B). These results suggest that the structure/stability rather than sequence of the predicted stem, other than G at +1 position determines the level of RNA editing.

To further examine the importance of the stability and structure of the stem, we analyzed the A3G RNA substrate, *PRPSAP2* and the A3A substrate *TMEM109.* Interestingly, weakening the putative stem by substituting two G-C base pairs with A-U base pairs in *PRPSAP2* RNA only affected RNA editing slightly (80%) (Fig. 1D, M6P). As mentioned above, disrupting the predicted stem structure abolished RNA editing in *PRPSAP2* (Fig. 1D, M5P). However, on swapping the 5′ and 3′ sequence while maintaining the stem complementarity as well as the first C-G base pair, increased RNA editing to 180% as compared to WT *PRPSAP2* (Fig. 1D, M7P). Similarly, when we compare the *SDHB* RNA mutants M6 and M10 (Fig. 1B), restoring the stem stability and structure, while maintaining G at +1 position increased RNA editing from 12% to 130%. These results provide further evidence that stem stability and G at +1 position, rather than nucleotide sequence in the remainder of the predicted stem region determine the level of RNA editing.

As mentioned above, for the A3A substrate *TMEM109,* the Mfold program predicts a hepta-loop flanked by a four bp stem (Fig. S1). However, if the unpaired adenosine in the hepta-loop region bulges out then we predict WT *TMEM109* RNA to form a tetra-loop with C at the 3′ end of the loop, G at +1 position base paired with C and a 5 bp long stem (Fig. 1C). To test the effect of perfect stem complementarity on *TMEM109* RNA editing level, we deleted the unpaired adenosine (Fig. 1C, M1T). Unlike for WT *TMEM109* ($\Delta G = -1.7$ kcal/mol), the Mfold program predicts a $\Delta G$ value of $-5.2$ kcal/mol for *TMEM109* M1T structure (Fig. S1), suggesting an increase in secondary structure stability. Deletion of the unpaired adenosine to obtain perfect stem complementarity resulted in an increase in the RNA editing level of *TMEM109* from 100% to 122% (Fig. 1C, M1T).

Taken together, our results show that for site-specific RNA editing, A3A and A3G prefer a stem-loop secondary structure, with C at the end of the tetra-loop as well as specific nucleotides at 5′ and 3′ positions immediate to the reactive C , and suggests that the sequence of the predicted stem other than at +1 position is not as important as the stability of base pairing.

## DISCUSSION

Most of the structural and biochemical studies of A3A and A3G thus far have focused on ssDNA substrate binding and the mechanism of catalysis. Moreover, it has been suggested

that RNA is not a substrate for A3A and A3G (*Iwatani et al., 2006*; *Mitra et al., 2014*). This is primarily because prior studies have shown DNA editing whereas RNA editing by the APOBEC3 enzymes was not observed until we demonstrated the RNA editing function of A3A and A3G recently (*Sharma et al., 2015*; *Sharma et al., 2016*; *Sharma et al., 2017*). The observation that RNA editing is site-specific with edited NNNC flanked by inverted repeats, whereas DNA editing occurs non-specifically at dinucleotide [T/C]C sequences motivated us to investigate the RNA secondary structure preference for A3A and A3G. Here, we show that stem-loop structures, with the reactive C contained in the loop, are preferred substrates for site-specific A3A and A3G-mediated RNA editing (Fig. 1).

Our results suggest that the determinants of RNA editing lie within the predicted loop of the stem loop structure, the +1 nucleotide in the stem, while the level of editing may be determined by the stem stability. Changing TC>CC in the *SDHB* RNA (A3A substrate) and changing CC>GC in *PRPSAP2* RNA (A3G substrate) markedly reduces or abolishes RNA editing by these enzymes respectively, thus highlighting the importance of the −1 nt in the loop (Fig. 1B, M1 and Fig. 1D, M1P). Another important feature is the +1 nucleotide (G) located in the putative stem common to all substrates of A3A and A3G examined here. Any substitution of G at the +1 position in these substrates markedly reduces RNA editing (Fig. 1B, M8, M9). In contrast to a predicted tetra-loop or a tri-loop, a predicted penta-loop RNA shows poor editing by both A3A and A3G (Fig. 1B, M3 and Fig. 1D, M2P). This may be because the catalytic site of these proteins is not 'open' or flexible enough to accommodate the larger RNA loop or because C is not present at the end of the loop in these mutants. The level of RNA editing by A3A and A3G in *SDHB* and *PRPSAP2* RNAs, respectively increases when compared to WT when the predicted stem sequence is altered while retaining the first base pair and the stem stability (Fig. 1B, M10 and Fig. 1D, M7P) or if we increase the stem stability of the A3A substrate *TMEM109* RNA by deleting the unpaired adenosine (Fig. 1C, M1T). These mutations may result in a more energetically favourable secondary structure for RNA editing or may result in a better 'fit' and interaction of the bases with the catalytic and surrounding residues.

Secondary structures of RNAs have been previously shown to aid in site-specific editing by adenosine deaminases in both prokaryotes and eukaryotes. The adenosine deaminases, ADARs, act on double stranded RNAs (dsRNAs) to convert adenosine to inosine. Secondary structure in the form of internal loops, bulges and mismatches in the dsRNAs dictate site-specificity in these enzymes resulting in the editing of a few adenosines as compared with long (>100 bp) dsRNA substrates, in which more than half of the adenosines are edited (*Lehmann & Bass, 1999*; *Bass, 2002*; *Nishikura, 2016*; *Deffit & Hundley, 2016*). The site selectivity in the glutamate receptor GRIA2, catalyzed by ADAR2, requires a stem structure that is formed between the exonic sequence containing the target A and a downstream intronic complementary sequence, resulting in >99% editing efficiency (*Higuchi et al., 1993*). Although ADARs prefer U at −1 and G at +1 position relative to the edited A, there is no strict sequence requirement for A>I editing (*Lehmann & Bass, 2000*; *Nishikura, 2016*). Also, the mechanism which determines the level of A>I RNA editing is not yet clear (*Nishikura, 2016*).

A distant relative of APOBECs, the prokaryotic adenosine deaminase TadA (Adenosine deaminase acting on tRNA or ADAT) has the active site characteristic of the cytidine deaminases and its mechanism of reaction is analogous to that of cytidine deaminases (*Carter Jr, 1995*; *Losey, Ruthenburg & Verdine, 2006*). TadA deaminates adenosine to inosine at the wobble position ($A^{34}$) of the tRNA$^{Arg2}$ anticodon stem-loop and involves an induced fit of the RNA stem-loop into an inflexible protein cleft (*Losey, Ruthenburg & Verdine, 2006*). Site-specific editing by TadA in the anticodon stem loop is achieved via its interactions with the loop and the single proximal base-pair of the stem, while the remainder of the stem participates in non-specific interactions with the protein, and the reactive adenosine lies within the deepest pocket on the enzyme (*Losey, Ruthenburg & Verdine, 2006*). Further, mutagenesis studies of the tRNA$^{Arg2}$ anti-codon stem-loop suggested the importance of the −1 nt, the size of the loop and structure of the stem as determinants of editing by TadA (*Wolf, Gerber & Keller, 2002*). Another example for secondary structure requirement for cytidine deamination is the Activation-induced cytidine deaminase (AID), which edits C nucleotides located within transcription bubbles or stem-loop structures in ssDNAs, independent of their sequence (*Larijani et al., 2007*). Recently, the crystal structure of A3A in complex with ssDNA 15-mer shows the DNA oligonucleotide adopting a bent conformation with C inserted in the active site of A3A (*Kouno et al., 2017*). A crystal structure of WT A3A/A3G in complex with its ssRNA substrate is crucial to understand the mechanism of protein-RNA interaction and catalysis.

The novel implication of our work is the effect of single nucleotide polymorphisms (SNPs) on the level of RNA editing. The G allele of a common A/G synonymous SNP in *C1QA* (rs172378) has been previously linked to an increased risk of disease severity and nephritis in systemic lupus erythematosus (*Namjou et al., 2009*; *Radanova et al., 2015*). We observed that this SNP increases the level of site-specific C>U RNA editing three nucleotides upstream of the polymorphism in primary monocytes exposed to hypoxia and interferons (Fig. 1A). RNA editing levels are 11% and 21% in two A/A homozygous donors but are increased to 40% in an A/G heterozygous donor (Fig. 1A). Although *C1QA* RNA editing at this site does not change the amino acid (CUC>CUU, both coding for leucine), our findings provide evidence that the G allele of rs172378 may alter the secondary structure of mRNA to favor a stronger stem and thereby increase the RNA editing level. This alteration in the predicted stem-loop structure may in turn affect mRNA stability, turnover or translatability (*Nackley et al., 2006*). Furthermore, it is conceivable that certain synonymous SNPs could create protein diversity by regulating the level of RNA editing. Few examples from our mutagenesis studies include substitutions in the *SDHB* DNA template (Fig. 1B, M1 and M3), where we changed −1T>C (Y45Y) and −4C>T (I44I). Although these mutations are synonymous, they markedly reduce the level of c.136 C>U RNA editing, which causes R46X alteration in *SDHB* RNA. Similarly, on making synonymous substitutions in the A3G substrate *PRPSAP2* by changing −1C>G (CCcC>CCgC; P267P) and −4C>G (GCcCCCC>GCgCCCC; A266A) (mutated residue in lower case) (Fig. 1D, M1P and M2P, respectively), there is a drastic reduction in RNA editing (CGG>UGG; R268W) that causes a missense alteration in *PRPSAP2*. Mutations in the *APP* gene have been linked to Alzheimer's disease. When we change −1C>T (CTcCGU>CTtC GU; L515L),

this synonymous mutation increases the editing level of the missense RNA alteration (CGU>UGU; R516C) by 264% (Fig. 1C, M1A). Thus, synonymous SNPs in the vicinity of the target C could alter expression of the translated product by regulating the levels of site-specific recoding C>U RNA editing.

## CONCLUSIONS

RNA editing is a mechanism to diversify information encoded by a gene and of regulation of gene expression. Our work provides the first experimental information on how stem-loop structures of endogenous RNA substrates may be preferred for site-specific editing mediated by A3A and A3G cytidine deaminases that are highly expressed in innate immune cells. These enzymes have hundreds of substrates and a single synonymous mutation altering the secondary structure in the substrate RNA could have consequences on the resulting protein product. It is possible that other APOBEC3 enzymes may prefer stem-loop structures, pending the discovery of their RNA editing function. Thus, this study provides the basis for future structural and functional studies.

## ACKNOWLEDGEMENTS

We thank Paul Gollnick for critical reading of the manuscript and for his suggestions. We thank Sally M. Enriquez for purification of the C-terminal his tagged WT A3A protein.

### Funding

This work was supported by National Cancer Institute (NCI) (P30CA016056) involving the use of Roswell Park Cancer Institute (RPCI)'s Genomics Shared Resources and RPCI Startup funds. The funders had no role in study design, data collection and analysis, decision to publish, or preparation of the manuscript.

### Grant Disclosures

The following grant information was disclosed by the authors:
National Cancer Institute (NCI): P30CA016056.
RPCI Startup funds.

### Competing Interests

The authors declare there are no competing interests.

### Author Contributions

- Shraddha Sharma conceived and designed the experiments, performed the experiments, analyzed the data, contributed reagents/materials/analysis tools, wrote the paper, prepared figures and/or tables, reviewed drafts of the paper.
- Bora E. Baysal conceived and designed the experiments, analyzed the data, contributed reagents/materials/analysis tools, wrote the paper, reviewed drafts of the paper.

## Data Availability

The raw data has been provided as Supplemental Files.

## Supplemental Information

Supplemental information for this article can be found online at http://dx.doi.org/10.7717/peerj.4136#supplemental-information.

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
