# Peer review of "Stem-loop structure preference for site-specific RNA editing by APOBEC3A and APOBEC3G"

_PeerJ, doi:10.7717/peerj.4136_

## Round 0.1 · original submission · Minor Revisions

· Academic Editor

Minor Revisions

Stem-loop structure preference for site-specific RNA editing by APOBEC3A and APOBEC3G
PeerJ 19608

Dear Drs Baysal and Sharma,

Thank you for giving us the opportunity to consider your manuscript, which has been read and critiqued by two expert referees.

The referees' comments were generally positive and, in principle, we would be pleased to publish your work provided you respond appropriately to the Reviewers’ concerns. Reviewer 1 would prefer broader citation of references and broader Discussion, but raises no major concerns.

Reviewer 2 has minor concerns similar to Reviewer 1, but also raised more serious concerns about data analysis and presentation that should be addressed. In particular, background base-call peaks comparable in size to C>U “editing” peaks are visible in samples 3N and 3HI in Figure 1A, which raised concerns about discerning low levels of deamination from background noise. I am further concerned that the normalisation of editing in w.t. RNA substrates to 100% obscures the dynamic range of the assays used. These concerns might be addressed by including the use of A3A and A3G catalytic mutants in the assays as suggested by Reviewer 2, by including raw data in Figures 1B-D, or by other means the authors deem suitable.

Yours sincerely,
Chris Jolly, PhD

·

Basic reporting

In this manuscript, Sharma et al. have examined the role of stemloop structure and sequence in A3A/G activity. The work follows up on previously published findings by the group that A3A can edit RNA in monocytes and macrophages. In this manuscript, the impact of RNA shape and sequence is examined.

Experimental design

I found experimental design to be quite standard and sound. A3A can be purified from eukaryotic or prokaryotic expression systems, and in the present work the authors have used a C-terminally His-tagged version purified form a bacterial expression system. This protein is incubated with the RNA substrates, mutations are measured by sequencing. Overall, the protocol follows established assays and data analysis was sufficiently detailed and rigorous.

Validity of the findings

The study identifies important structure and sequence aspects of the RNA substrates. The data is in line with the well-established sensitivity of all AID/APOBEC family member enzymes to ssDNA shapes and sequence motifs. The functional data correlated nicely with predicted RNA secondary structures in a manner expected from the known biochemical properties of AID/APOBEC family member enzymes. I did not find grounds to doubt the validity of the presented findings.

Additional comments

In general, I found the manuscript was very well put together, clear and easy to follow. The conclusions are supported by the data and the work was rigorously carried out. My suggestion to the authors are to include additional more comprehensive references:

-For instance, in regards to the role of A3s in viral restriction, there is an emerging body of evidence that A3s can also do the opposite, by helping retroviruses escape from drugs and adaptive immune recognition (suggested reference Monajemi et al. 2012, Retrovirology)

-In regards to the structure/function of A3 catalytic motifs as discussed in the introduction, there are some nice current structural studies focusing on the catalytic pockets of AID/APOBECs which can be referenced (e.g. King et al. 2017 Frontiers Imm, King et al. 2015 Structure, Shi et al. 2017 NSMB)

-The discussion will be improved if the authors also discuss their work in the context of previous works on the relative importance of secondary structure vs. primary sequence for determining AID/APOBECs. For instance, Larijani et al. 2007 MCB (amongst others) did similar work on AID and found that secondary structure (shape) is a more important determinant of AID targeting to stemloops and bubbles, as compared to its trinucleotide WRC sequence specificity.

Reviewer 2 ·

Basic reporting

Very minor comment on the references: it is a bit strange that introduction starts by citing reviews, which is perfectly fine considering the time passed from the initial characterisation of the APOBEC3s, yet only one review of the many labs involved in the initial characterisation (with a few still working on the APOBEC3 genes) is used.
Then the introduction moves to describe the APOBEC3 locus, going back to an original paper (Jarmuz 2002 - incidentally the organisation A3 locus described in that paper is different from the current understanding), but Jarmuz 2002 is not the proper reference for the Zn-coordinating residues in the deaminase domain (it should be Betts 1994).

Experimental design

no comments

Validity of the findings

The findings support the conclusions, pending the clarification of the problems highlighted below:
a) lack of negative controls
- With regards to the experiments with patients' MEPs, it would be useful to also have a sample in which APOBEC3A is silenced (the same authors have used siRNAs in the past). In Figure 1 there are background peaks in the chromatograms from Donor 2 and 3 HI that are the same height as the edited peak. It seems that even the edited peak could be considered background (which does not interfere with the conclusions)
- With regards to the in vitro experiments and the A3G ones, it would be necessary to use a catalytically inactive mutant of the enzyme to show the background levels of editing.
All this is necessary as it is not clear what is the dynamic range and the linearity of the editing assays.
b) It is not specified how many times the experiments were repeated, and there is no sign of error bars (or statistics) in any of the experiments.
c) in parts of Fig.1 the editing of the mutants is quantified through different methods (Sanger sequencing, AS-RT-qPCR). How do these assays compare with each other? Could it be better to keep the experiments separated?

Additional comments

The topic is fascinating and the findings have the potential to help us understand the dynamics of APOBEC-mediated RNA editing.
The striking observation that specificity and strength of RNA editing seems to be deriving from a segment of 10-16 nucleotides is somehow at odds with the specificity of editing previously described. How can such specificity be conferred by a few bases in the loop? What other elements are necessary for other the editing to be targeted to those transcripts?
Considering the data the Authors have accumulated in these years, it might be worth reanalysing the RNA-seq data and seek whether other transcribed sequences bearing identical stem-loop segments lack some other feature present in the edited transcripts.

---

## Round 0.2 · Minor Revisions

· Academic Editor

Minor Revisions

Dear Drs Baysal and Sharma,

Thank you again for giving us the opportunity to consider your manuscript, which has been re-read and critiqued by referee 2 and myself. I apologize for the delays in reviewing on behalf of Referee 2 and myself. 

You have addressed all of our concerns – most particularly by clarifying that the appearance of an edited U/T-peak is accompanied by a reduction in the precursor C-peak in the Fig.1 Legend. I apologize if that clarification was present in the Legend in the original version and missed by Reviewer 2 and myself.

Thank you also for including all raw data in the additional supplemental files. However, I cannot find any reference to these new Supplemental files in the manuscript’s text. I request that you make the very minor revision of adding references to these files at suitable places in the Results section. I also ask that you rename Supplementary Figure 2 as Supplementary Data, since it is not really a Figure, then the manuscript should be ready for publication in my opinion.

Your discovery of RNA editing by APOBEC3 deaminases is exciting, and this paper makes an important and thorough contribution to the biology of nucleic acid deamination. Thank you for considering PeerJ.

Yours sincerely,
Chris Jolly, PhD

Reviewer 2 ·

Basic reporting

no comment

Experimental design

no comment

Validity of the findings

no comment

Additional comments

The rebuttal of the Authors answers most of my criticism. Yet I wonder why none of their argumentations end up in the manuscript. I would have thought some of those answers would provide context to the presented work and deserve a place in the study.

Specific comments:
- as I have already written, the background peaks in Donor 2 and 3 do not change the interpretation of the phenomenon, but they do cast doubts on the background levels of the assay and therefore on its dynamic range. In short, if you want to use a live cell assay to study biochemistry, you have to define the dynamic range of your assay, otherwise all your observations are just qualitative (which might be fine, but it should be noted). This is something the Authors should deal with.

- I apologize for missing the number of times the experiments were repeated (though it is not the usual notation) and the standard deviation. Would it be meaningful to add some statistics to describe the various efficiencies?

- Do the authors realize that maybe not every reviewer/reader is familiar with Supplementary Fig. 8 of Sharma 2015? It might be helpful a citation in the text to support the correlation between the two techniques. Moreover the slope in Suppl. Fig 8 is 0.7. Does this mean that the values were normalized between experiments in order to compare the two techniques? Would it be useful to label each substrate to highlight which experiments the data came from?

- Actually (aside from correcting the Betts reference), the references were just fine. It was just odd that the only original work cited at the beginning was the Jarmuz paper, whose importance with regards to the research presented is marginal if compared to other original research (e.g Sheehy 2002, Harris 2003, Mangeat 2003, etc.). It was odd, but a legitimate choice.
On the other hand, if Jarmuz is there because it is the first reference for A3A, then it would be fair to also cite the original Madsen 1999 [phorbolin] and Sheehy 2002 [identification of A3G] papers.

---

## Round 0.3 · accepted · Accept

· Academic Editor

Accept

The topic of RNA editing by APOBEC deaminases is a very exciting one, and this paper makes an important and thorough contribution. Thank you for considering PeerJ.